# A burden of proof study on alcohol consumption and ischemic heart disease

Sinclair Carr [1] ✉, Dana Bryazka[1], Susan A. McLaughlin[1], Peng Zheng[1,2], Sarasvati Bahadursingh[3], Aleksandr Y. Aravkin[1,2,4], Simon I. Hay [1,2], Hilary R. Lawlor[1], Erin C. Mullany[1], Christopher J. L. Murray [1,2], Sneha I. Nicholson[1], Jürgen Rehm[5,6,7,8,9,10,11,12], Gregory A. Roth[1,2,13], Reed J. D. Sorensen[1], Sarah Lewington[3] & Emmanuela Gakidou [1,2]

Cohort and case-control data have suggested an association between low to moderate alcohol consumption and decreased risk of ischemic heart disease (IHD), yet results from Mendelian randomization (MR) studies designed to reduce bias have shown either no or a harmful association. Here we conducted an updated systematic review and re-evaluated existing cohort, case-control, and MR data using the burden of proof meta-analytical framework. Cohort and case-control data show low to moderate alcohol consumption is associated with decreased IHD risk – specifically, intake is inversely related to IHD and myocardial infarction morbidity in both sexes and IHD mortality in males – while pooled MR data show no association, confirming that self-reported versus genetically predicted alcohol use data yield conflicting findings about the alcohol-IHD relationship. Our results highlight the need to advance MR methodologies and emulate randomized trials using large observational databases to obtain more definitive answers to this critical public health question.

It is well known that alcohol consumption increases the risk of morbidity and mortality due to many health conditions[1,2], with even low levels of consumption increasing the risk for some cancers[3,4]. In contrast, a large body of research has suggested that low to moderate alcohol intake – compared to no consumption – is associated with a decreased risk of ischemic heart disease (IHD). This has led to substantial epidemiologic and public health interest in the alcohol-IHD relationship[5], particularly given the high prevalence of alcohol consumption[6] and the global burden of IHD[7].

Extensive evidence from experimental studies that vary short-term alcohol exposure suggests that average levels of alcohol intake positively affect biomarkers such as apolipoprotein A1, adiponectin, and fibrinogen levels that lower the risk of IHD[8]. In contrast, heavy episodic drinking (HED) may have an adverse effect on IHD by affecting blood lipids, promoting coagulation and thus thrombosis risk, and increasing blood pressure[9]. With effects likely to vary materially by patterns of drinking, alcohol consumption must be considered a multidimensional factor impacting IHD outcomes.

[1]Institute for Health Metrics and Evaluation, University of Washington, Seattle, WA, USA. [2]Department of Health Metrics Sciences, School of Medicine, University of Washington, Seattle, WA, USA. [3]Clinical Trial Service Unit & Epidemiological Studies Unit, Nuffield Department of Population Health, University of Oxford, Oxford, Oxfordshire, UK. [4]Department of Applied Mathematics, University of Washington, Seattle, WA, USA. [5]Institute for Mental Health Policy Research, Centre for Addiction and Mental Health, Toronto, ON, Canada. [6]Campbell Family Mental Health Research Institute, Centre for Addiction and Mental Health, Toronto, ON, Canada. [7]Dalla Lana School of Public Health, University of Toronto, Toronto, ON, Canada. [8]Department of Psychiatry, University of Toronto, Toronto, ON, Canada. [9]Faculty of Medicine, Institute of Medical Science (IMS), University of Toronto, Toronto, ON, Canada. [10]World Health Organization / Pan American Health Organization Collaborating Centre, Centre for Addiction and Mental Health, Toronto, ON, Canada. [11]Center for Interdisciplinary Addiction Research (ZIS), Department of Psychiatry and Psychotherapy, University Medical Center Hamburg-Eppendorf (UKE), Hamburg, Germany. [12]Institute of Clinical Psychology and Psychotherapy, Technische Universität Dresden, Dresden, Germany. [13]Division of Cardiology, Department of Medicine, University of Washington, Seattle, WA, USA. ✉e-mail: scarr@hsph.harvard.edu

A recent meta-analysis of the alcohol-IHD relationship using individual participant data from 83 observational studies[4] found, among current drinkers, that – relative to drinking less than 50 g/week – any consumption above this level was associated with a lower risk of myocardial infarction (MI) incidence and consumption between >50 and <100 g/week was associated with lower risk of MI mortality. When evaluating other subtypes of IHD excluding MI, the researchers found that consumption between >100 and <250 g/week was associated with a decreased risk of IHD incidence, whereas consumption greater than 350 g/week was associated with an increased risk of IHD mortality. Roerecke and Rehm further observed that low to moderate drinking was not associated with reduced IHD risk when accompanied by occasional HED[10].

The cohort studies and case-control studies (hereafter referred to as 'conventional observational studies') used in these meta-analyses are known to be subject to various types of bias when used to estimate causal relationships[11]. First, neglecting to separate lifetime abstainers from former drinkers, some of whom may have quit due to developing preclinical symptoms (sometimes labeled 'sick quitters'[12,13]), and to account for drinkers who reduce their intake as a result of such symptoms may introduce reverse causation bias[13]. That is, the risk of IHD in, for example, individuals with low to moderate alcohol consumption may be lower when compared to IHD risk in sick quitters, not necessarily because intake at this level causes a reduction in risk but because sick quitters are at higher risk of IHD. Second, estimates can be biased because of measurement error in alcohol exposure resulting from inaccurate reporting, random fluctuation in consumption over time (random error), or intentional misreporting of consumption due, for example, to social desirability effects[14] (systematic error). Third, residual confounding may bias estimates if confounders of the alcohol-IHD relationship, such as diet or physical activity, have not been measured accurately (e.g., only via a self-report questionnaire) or accounted for. Fourth, because alcohol intake is a time-varying exposure, time-varying confounding affected by prior exposure must be accounted for[15]. To date, only one study that used a marginal structural model to appropriately adjust for time-varying confounding found no association between alcohol consumption and MI risk[16]. Lastly, if exposure to a risk factor, such as alcohol consumption, did not happen at random – even if all known confounders of the relationship between alcohol and IHD were perfectly measured and accounted for – the potential for unmeasured confounders persists and may bias estimates[11].

In recent years, the analytic method of Mendelian randomization (MR) has been widely adopted to quantify the causal effects of risk factors on health outcomes[17–19]. MR uses single nucleotide polymorphisms (SNPs) as instrumental variables (IVs) for the exposure of interest. A valid IV should fulfill the following three assumptions: it must be associated with the risk factor (relevance assumption); there must be no common causes of the IV and the outcome (independence assumption); and the IV must affect the outcome only through the exposure (exclusion restriction or 'no horizontal pleiotropy' assumption)[20,21]. If all three assumptions are fulfilled, estimates derived from MR are presumed to represent causal effects[22]. Several MR studies have quantified the association between alcohol consumption and cardiovascular disease[23], including IHD, using genes known to impact alcohol metabolism (e.g., ADH1B/C and ALDH2[24]) or SNP combinations from genome-wide association studies[25]. In contrast to the inverse associations found in conventional observational studies, MR studies have found either no association or a harmful relationship between alcohol consumption and IHD[26–31].

To advance the knowledge base underlying our understanding of this major health issue – critical given the worldwide ubiquity of alcohol use and of IHD – there is a need to systematically review and critically re-evaluate all available evidence on the relationship between alcohol consumption and IHD risk from both conventional observational and MR studies.

The burden of proof approach, developed by Zheng et al.[32], is a six-step meta-analysis framework that provides conservative estimates and interpretations of risk-outcome relationships. The approach systematically tests and adjusts for common sources of bias defined according to the Grading of Recommendations Assessment, Development, and Evaluation (GRADE) criteria: representativeness of the study population, exposure assessment, outcome ascertainment, reverse causation, control for confounding, and selection bias. The key statistical tool to implement the approach is MR-BRT (meta-regression −Bayesian, regularized, trimmed[33]), a flexible meta-regression tool that does not impose a log-linear relationship between the risk and outcome, but instead uses a spline ensemble to model non-linear relationships. MR-BRT also algorithmically detects and trims outliers in the input data, takes into account different reference and alternative exposure intervals in the data, and incorporates unexplained between-study heterogeneity in the uncertainty surrounding the mean relative risk (RR) curve (henceforth 'risk curve'). For those risk-outcome relationships that meet the condition of statistical significance using conventionally estimated uncertainty intervals (i.e., without incorporating unexplained between-study heterogeneity), the burden of proof risk function (BPRF) is derived by calculating the 5th (if harmful) or 95th (if protective) quantile risk curve – inclusive of between-study heterogeneity – closest to the log RR of 0. The resulting BPRF is a conservative interpretation of the risk-outcome relationship based on all available evidence. The BPRF represents the smallest level of excess risk for a harmful risk factor or reduced risk for a protective risk factor that is consistent with the data, accounting for between-study heterogeneity. To quantify the strength of the evidence for the alcohol-IHD relationship, the BPRF can be summarized in a single metric, the risk-outcome score (ROS). The ROS is defined as the signed value of the average log RR of the BPRF across the 15th to 85th percentiles of alcohol consumption levels observed across available studies. The larger a positive ROS value, the stronger the alcohol-IHD association. For ease of interpretation, the ROS is converted into a star rating from one to five. A one-star rating (ROS < 0) indicates a weak alcohol-IHD relationship, and a five-star rating (ROS > 0.62) indicates a large effect size and strong evidence. Publication and reporting bias are evaluated with Egger's regression and by visual inspection with funnel plots[34]. Further conceptual and technical details of the burden of proof approach are described in detail elsewhere[32].

Using the burden of proof approach, we systematically re-evaluate all available eligible evidence from cohort, case-control, and MR studies published between 1970 and 2021 to conservatively quantify the dose-response relationship between alcohol consumption and IHD risk, calculated relative to risk at zero alcohol intake (i.e., current non-drinking, including lifetime abstinence or former use). We pool the evidence from all conventional observational studies combined, as well as individually for all three study designs, to estimate mean IHD risk curves. Based on patterns of results established by previous meta-analyses[4,35], we also use data from conventional observational studies to estimate risk curves by IHD endpoint (morbidity or mortality) and further by sex, in addition to estimating risk curves for MI overall and by endpoint. We follow PRISMA (Preferred Reporting Items for Systematic Reviews and Meta-Analyses) guidelines[36] through all stages of this study (Supplementary Information section 1, Fig. S1 and Tables S1 and S2) and comply with GATHER (Guidelines on Accurate and Transparent Health Estimates Reporting) recommendations[37] (Supplementary Information section 2, Table S3). The main findings and research implications of this work are summarized in Table 1.

## Results

We updated the systematic review on the dose-response relationship between alcohol consumption and IHD previously conducted for the Global Burden of Diseases, Injuries, and Risk Factors Study (GBD)

**Table 1 | Research summary**

| | |
|---|---|
| Background | Different study designs have yielded conflicting evidence about whether low to moderate levels of alcohol consumption are associated with increased or decreased risk of ischemic heart disease (IHD), a leading cause of ill health and death worldwide. Results from cohort and case-control studies predominantly show that average consumption is associated with decreased IHD risk, although findings vary by sex, disease endpoint (morbidity versus mortality), and engagement in heavy episodic drinking (HED). Conversely, Mendelian randomization (MR) studies relying on genetic variants that predict alcohol use typically find no association or harmful relationship between alcohol and IHD. |
| Main findings and limitations | Cautious re-evaluation using the burden of proof meta-analytic methods – systematically applied to capture potential non-log-linear relationships, control for known sources of bias, and incorporate explained and unexplained between-study heterogeneity to generate conservative estimates of risk-outcome associations – yielded estimates of the alcohol-IHD relationship that varied by study design, consistent with previous findings. Data pooled from cohort and case-control studies showed a weak association between average levels of alcohol consumption (up to ~50 g/day) and reduced IHD risk relative to no alcohol intake, while data pooled from MR studies showed no association between genetically predicted alcohol consumption and IHD risk. A primary limitation of the analysis is that it was not possible, due to insufficient data, to differentiate the estimated alcohol-IHD relationship by alcohol use subtype – that is, by average consumption characterized by frequency and quantity, by HED, or by beverage type – and to then compare subtype-specific relationships across study designs. |
| Implications | Using a conservative approach to consider all evidence from cohort, case-control, and MR studies, we confirmed conflicting estimates of the relationship between alcohol use and IHD derived from self-reports of intake levels versus genetically predicted alcohol use. The discrepant findings are likely driven by biases and limitations inherent in the different study designs and highlight the need to advance methodologies to obtain more definitive answers to this critical public health question. The rapidly evolving field of MR makes it possible to apply new, sophisticated MR techniques to mitigate the effects of bias in investigations of this relationship. Long-term randomized trials can be emulated using large observational databases, avoiding some of the limitations common to conventional observational studies. New MR and trial emulation approaches should be considered as ways forward to more conclusively answer pressing questions about the potential effects of alcohol consumption on IHD. <br><br> It is anticipated that the present synthesis of evidence from cohort, case-control, and MR studies assessing the dose-response relationship between alcohol intake and IHD risk will be incorporated in upcoming iterations of the Global Burden of Diseases, Injuries, and Risk Factors Study (GBD). |

2020[1]. Of 4826 records identified in our updated systematic review (4769 from databases/registers and 57 by citation search and known literature), 11 were eligible based on our inclusion criteria and were included. In total, combined with the results of the previous systematic reviews[1,38], information from 95 cohort studies[26,27,29,39–130], 27 case-control studies[131–157], and five MR studies[26–29,31] was included in our meta-analysis (see Supplementary Information section 1, Fig. S1, for the PRISMA diagram). Details on the extracted effect sizes, the design of each included study, underlying data sources, number of participants, duration of follow-up, number of cases and controls, and bias covariates that were evaluated and potentially adjusted for can be found in the Supplementary Information Sections 4, 5, and 6.

Table 2 summarizes key metrics of each risk curve modeled, including estimates of mean RR and 95% UI (inclusive of between-study heterogeneity) at select alcohol exposure levels, the exposure level and RR and 95% UI at the nadir (i.e., lowest RR), the 85th percentile of exposure observed in the data and its corresponding RR and 95% UI, the BPRF averaged at the 15th and 85th percentile of exposure, the average excess risk or risk reduction according to the exposure-averaged BPRF, the ROS, the associated star rating, the potential presence of publication or reporting bias, and the number of studies included.

We found large variation in the association between alcohol consumption and IHD by study design. When we pooled the results of cohort and case-control studies, we observed an inverse association between alcohol at average consumption levels and IHD risk; that is, drinking average levels of alcohol was associated with a reduced IHD risk relative to drinking no alcohol. In contrast, we did not find a statistically significant association between alcohol consumption and IHD risk when pooling results from MR studies. When we subset the conventional observational studies to those reporting on IHD by endpoint, we found no association between alcohol consumption and IHD morbidity or mortality due to large unexplained heterogeneity between studies. When we further subset those studies that reported effect size estimates by sex, we found that average alcohol consumption levels were inversely associated with IHD morbidity in males and in females, and with IHD mortality in males but not in females. When we analyzed only the studies that reported on MI, we found significant inverse associations between average consumption levels and MI

overall and with MI morbidity. Visualizations of the risk curves for morbidity and mortality of IHD and MI are provided in Supplementary Information Section 9 (Figs. S2a–c, S3a–c, and S4a–c). Among all modeled risk curves for which a BPRF was calculated, the ROS ranged from −0.40 for MI mortality to 0.20 for MI morbidity. In the Supplementary Information, we also provide details on the RR and 95% UIs with and without between-study heterogeneity associated with each 10 g/day increase in consumption for each risk curve (Table S10), the parameter specifications of the model (Tables S11 and S12), and each risk curve from the main analysis estimated without trimming 10% of the data (Fig. S5a–l and Table S13).

**Risk curve derived from conventional observational study data**
The mean risk curve and 95% UI were first estimated by combining all evidence from eligible cohort and case-control studies that quantified the association between alcohol consumption and IHD risk. In total, information from 95 cohort studies and 27 case-control studies combining data from 7,059,652 participants were included. In total, 243,357 IHD events were recorded. Thirty-seven studies quantified the association between alcohol consumption and IHD morbidity only, and 44 studies evaluated only IHD mortality. The estimated alcohol-IHD association was adjusted for sex and age in all but one study. Seventy-five studies adjusted the effect sizes for sex, age, smoking, and at least four other covariates. We adjusted our risk curve for whether the study sample was under or over 50 years of age, whether the study outcome was consistent with the definition of IHD (according to the International Classification of Diseases [ICD]−9: 410-414; and ICD-10: I20-I25) or related to specified subtypes of IHD, whether the outcome was ascertained by self-report only or by at least one other measurement method, whether the study accounted for risk for reverse causation, whether the reference group was non-drinkers (including lifetime abstainers and former drinkers), and whether effect sizes were adjusted (1) for sex, age, smoking, and at least four other variables, (2) for apolipoprotein A1, and (3) for cholesterol, as these bias covariates were identified as significant by our algorithm.

Pooling all data from cohort and case-control studies, we found that alcohol consumption was inversely associated with IHD risk (Fig. 1). The risk curve was J-shaped – without crossing the null RR of 1 at high exposure levels – with a nadir of 0.69 (95% UI: 0.48–1.01) at

**Table 2 | Strength of the evidence for the relationship between alcohol consumption and ischemic heart disease**

| | RR (95% UI) at select exposure levels | | | Nadir exposure level | RR (95% UI) at nadir | 85th percentile risk level | RR (95% UI) at 85th percentile risk level | Exposure-averaged BPRF | Conservative interpretation of the average risk increase/decrease | ROS | Star rating | Pub. bias | No. of studies |
|---|---|---|---|---|---|---|---|---|---|---|---|---|---|
| | 10 g/day | 30 g/day | 50 g/day | | | | | | | | | | |
| Ischemic heart disease | 0.76 (0.57, 1.01) | 0.70 (0.48, 1.01) | 0.73 (0.53, 1.01) | 23 g/day | 0.69 (0.48, 1.01) | 45 g/day | 0.72 (0.51, 1.01) | 0.96 | 4% | 0.04 | ★★ | Yes | 122 |
| Morbidity | 0.76 (0.51, 1.14) | 0.66 (0.36, 1.22) | 0.71 (0.42, 1.18) | 26 g/day | 0.66 (0.36, 1.22) | 45 g/day | 0.70 (0.41, 1.19) | 1.08 | N/A | −0.08 | ★ | No | 37 |
| Females | 0.77 (0.56, 1.04) | 0.81 (0.64, 1.03) | 0.86 (0.72, 1.02) | 15 g/day | 0.74 (0.53, 1.05) | 26 g/day | 0.79 (0.61, 1.04) | 0.99 | 1% | 0.01 | ★★ | No | 6 |
| Males | 0.63 (0.44, 0.91) | 0.58 (0.38, 0.89) | 0.73 (0.58, 0.93) | 20 g/day | 0.54 (0.33, 0.87) | 55 g/day | 0.76 (0.61, 0.94) | 0.86 | 14% | 0.15 | ★★★ | Yes | 6 |
| Mortality | 0.81 (0.55, 1.21) | 0.74 (0.42, 1.31) | 0.79 (0.51, 1.23) | 25 g/day | 0.74 (0.42, 1.32) | 50 g/day | 0.79 (0.51, 1.23) | 1.16 | N/A | −0.14 | ★ | Yes | 44 |
| Females | 0.86 (0.55, 1.33) | 0.83 (0.48, 1.42) | 0.86 (0.56, 1.32) | 20 g/day | 0.81 (0.44, 1.48) | 34 g/day | 0.84 (0.50, 1.39) | 1.25 | N/A | −0.22 | ★ | Yes | 8 |
| Males | 0.76 (0.64, 0.91) | 0.74 (0.61, 0.90) | 0.87 (0.80, 0.95) | 19 g/day | 0.70 (0.56, 0.88) | 58 g/day | 0.91 (0.85, 0.96) | 0.90 | 10% | 0.11 | ★★ | No | 8 |
| Case-control studies | 0.76 (0.64, 0.90) | 0.68 (0.53, 0.86) | 0.91 (0.85, 0.96) | 23 g/day | 0.65 (0.50, 0.85) | 45 g/day | 0.85 (0.77, 0.94) | 0.87 | 13% | 0.14 | ★★★ | No | 27 |
| Cohort studies | 0.76 (0.58, 1.00) | 0.69 (0.47, 1.01) | 0.72 (0.52, 1.00) | 23 g/day | 0.69 (0.47, 1.01) | 50 g/day | 0.72 (0.52, 1.00) | 0.95 | 5% | 0.05 | ★★ | Yes | 95 |
| Mendelian randomization studies | 1.00 (0.62, 1.61) | 1.00 (0.30, 3.38) | N/A | 0 g/day | N/A | 24 g/day | 1.00 (0.36, 2.81) | N/A | N/A | N/A | | No | 4 |
| Myocardial infarction | 0.68 (0.49, 0.97) | 0.69 (0.49, 0.97) | 0.73 (0.54, 0.97) | 15 g/day | 0.66 (0.45, 0.96) | 50 g/day | 0.73 (0.54, 0.97) | 0.92 | 8% | 0.08 | ★★ | No | 45 |
| Morbidity | 0.65 (0.49, 0.87) | 0.69 (0.54, 0.88) | 0.80 (0.68, 0.93) | 15 g/day | 0.62 (0.45, 0.85) | 49 g/day | 0.79 (0.68, 0.92) | 0.86 | 14% | 0.2 | ★★★ | Yes | 24 |
| Mortality | 0.86 (0.52, 1.41) | 0.76 (0.32, 1.83) | 0.75 (0.29, 1.92) | 84 g/day | 0.73 (0.26, 2.03) | 61 g/day | 0.74 (0.28, 1.96) | 1.50 | N/A | −0.4 | ★ | No | 9 |

The reported relative risk (RR) and its 95% uncertainty interval (UI) reflect the risk an individual who has been exposed to alcohol consumption has of developing ischemic heart disease or myocardial infarction relative to that of someone who does not drink alcohol (i.e., has zero intake). We report the 95% UI that incorporates unexplained between-study heterogeneity. The Burden of Proof Risk Function (BPRF) is calculated for risk-outcome pairs that were found to have significant relationships at a 0.05 level of significance when not incorporating between-study heterogeneity in the 95% UI. The BPRF corresponds to the 5th or 95th quantile estimate of relative risk accounting for between-study heterogeneity closest to the null for each relationship, and it reflects a conservative estimate of excess risk or risk reduction associated with alcohol consumption that is consistent with the available data. Since we define alcohol consumption as a continuous risk factor, the risk-outcome score (ROS) is calculated as the signed value of the log RR of the BPRF averaged between the 15th and 85th percentiles of exposure levels observed across studies. Negative ROSs indicate that the evidence of the association is weak and inconsistent. For ease of interpretation, we have transformed the ROS and BPRF into a star rating (1–5) with a higher rating representing a larger effect with stronger evidence. The potential existence of publication bias, which, if present, would affect the validity of the results, was tested using Egger's regression. Included studies represent all available relevant data identified through our systematic reviews from January 1970 through December 2021. N/A not available.

23 g/day. This means that compared to individuals who do not drink alcohol, the risk of IHD significantly decreases with increasing consumption up to 23 g/day, followed by a risk reduction that becomes less pronounced. The average BPRF calculated between 0 and 45 g/day of alcohol intake (the 15th and 85th percentiles of the exposure range observed in the data) was 0.96. Thus, when between-study heterogeneity is accounted for, a conservative interpretation of the evidence suggests drinking alcohol across the average intake range is associated with an average decrease in the risk of IHD of at least 4% compared to drinking no alcohol. This corresponds to a ROS of 0.04 and a star rating of two, which suggests that the association – on the basis of the available evidence – is weak. Although we algorithmically identified and trimmed 10% of the data to remove outliers, Egger's regression and visual inspection of the funnel plot still indicated potential publication or reporting bias.

### Risk curve derived from case-control study data

Next, we estimated the mean risk curve and 95% UI for the relationship between alcohol consumption and IHD by subsetting the data to case-control studies only. We included a total of 27 case-control studies (including one nested case-control study) with data from 60,914 participants involving 16,892 IHD cases from Europe ($n = 15$), North America ($n = 6$), Asia ($n = 4$), and Oceania ($n = 2$). Effect sizes were adjusted for sex and age in most studies ($n = 25$). Seventeen of these studies further adjusted for smoking and at least four other covariates. The majority of case-control studies accounted for the risk of reverse causation ($n = 25$). We did not adjust our risk curve for bias covariates, as our algorithm did not identify any as significant.

Evaluating only data from case-control studies, we observed a J-shaped relationship between alcohol consumption and IHD risk, with a nadir of 0.65 (0.50–0.85) at 23 g/day (Fig. 2). The inverse association between alcohol consumption and IHD risk reversed at an intake level of 61 g/day. In other words, alcohol consumption between >0 and 60 g/day was associated with a lower risk compared to no consumption, while consumption at higher levels was associated with increased IHD risk. However, the curve above this level is flat, implying that the association between alcohol and increased IHD risk is the same between 61 and 100 g/day, relative to not drinking any alcohol. The BPRF averaged across the exposure range between the 15th and 85th percentiles, or 0–45 g/day, was 0.87, which translates to a 13% average reduction in IHD risk across the average range of consumption. This corresponds to a ROS of 0.14 and a three-star rating. After trimming 10% of the data, no potential publication or reporting bias was found.

### Risk curve derived from cohort study data

We also estimated the mean risk curve and 95% UI for the relationship between alcohol consumption and IHD using only data from cohort studies. In total, 95 cohort studies – of which one was a retrospective cohort study – with data from 6,998,738 participants were included. Overall, 226,465 IHD events were recorded. Most data were from Europe ($n = 43$) and North America ($n = 33$), while a small number of studies were conducted in Asia ($n = 14$), Oceania ($n = 3$), and South America ($n = 2$). The majority of studies adjusted effect sizes for sex and age ($n = 76$). Fifty-seven of these studies also adjusted for smoking and at least four other covariates. Out of all cohort studies included, 88 accounted for the risk of reverse causation. We adjusted our risk curve for whether the study outcome was consistent with the definition of IHD or related to specified subtypes of IHD, and whether effect sizes were adjusted for apolipoprotein A1, as these bias covariates were identified as significant by our algorithm.

When only data from cohort studies were evaluated, we found a J-shaped relationship between alcohol consumption and IHD risk that did not cross the null RR of 1 at high exposure levels, with a nadir of 0.69 (0.47–1.01) at 23 g/day (Fig. 3). The shape of the risk curve was almost identical to the curve estimated with all conventional observational studies (i.e., cohort and case-control studies combined). When we calculated the average BPRF of 0.95 between the 15th and 85th percentiles of observed alcohol exposure (0–50 g/day), we found that alcohol consumption across the average intake range was associated with an average reduction in IHD risk of at least 5%. This corresponds to a ROS of 0.05 and a two-star rating. We identified potential publication or reporting bias after 10% of the data were trimmed.

### Risk curve derived from Mendelian randomization study data

Lastly, we pooled evidence on the relationship between genetically predicted alcohol consumption and IHD risk from MR studies. Four MR studies were considered eligible for inclusion in our main analysis, with data from 559,708 participants from China ($n = 2$), the Republic of Korea ($n = 1$), and the United Kingdom ($n = 1$). Overall, 22,134 IHD events were recorded. Three studies used the rs671 ALDH2 genotype found in Asian populations, one study additionally used the rs1229984 ADH1B variant, and one study used the rs1229984 ADH1B Arg47His variant and a combination of 25 SNPs as IVs. All studies used the two-stage least squares (2SLS) method to estimate the association, and one study additionally applied the inverse-variance-weighted (IVW) method and multivariable MR (MVMR). For the study that used multiple methods to estimate effect sizes, we used the 2SLS estimates for our main analysis. Further details on the included studies are provided in Supplementary Information section 4 (Table S6). Due to limited input data, we elected not to trim 10% of the observations. We adjusted our risk curve for whether the endpoint of the study outcome was mortality and whether the associations were adjusted for sex and/or age, as these bias covariates were identified as significant by our algorithm.

We did not find any significant association between genetically predicted alcohol consumption and IHD risk using data from MR studies (Fig. 4). No potential publication or reporting bias was detected.

As sensitivity analyses, we modeled risk curves with effect sizes estimated from data generated by Lankester et al.[28] using IVW and MVMR methods. We also used effect sizes from Biddinger et al.[31], obtained using non-linear MR with the residual method, instead of those from Lankester et al.[28] in our main model (both were estimated with UK Biobank data) to estimate a risk curve. Again, we did not find a significant association between genetically predicted alcohol consumption and IHD risk (see Supplementary Information Section 10, Fig. S6a–c and Table S14). To test for consistency with the risk curve we estimated using all included cohort studies, we also pooled the conventionally estimated effect sizes provided in the four MR studies. We did not observe an association between alcohol consumption and IHD risk due to large unexplained heterogeneity between studies (see Supplementary Information Section 10, Fig. S7, and Table S14). Lastly, we pooled cohort studies that included data from China, the Republic of Korea, and the United Kingdom to account for potential geographic influences. Again, we did not find a significant association between alcohol consumption and IHD risk (see Supplementary Information Section 10, Fig. S8, and Table S14).

## Discussion

Conventional observational and MR studies published to date provide conflicting estimates of the relationship between alcohol consumption and IHD. We conducted an updated systematic review and conservatively re-evaluated existing evidence on the alcohol-IHD relationship using the burden of proof approach. We synthesized evidence from cohort and case-control studies combined and separately and from MR studies to assess the dose-response relationship between alcohol consumption and IHD risk and to compare results across different study designs. It is anticipated that the present synthesis of evidence will be incorporated into upcoming iterations of GBD.

Our estimate of the association between genetically predicted alcohol consumption and IHD runs counter to our estimates from the

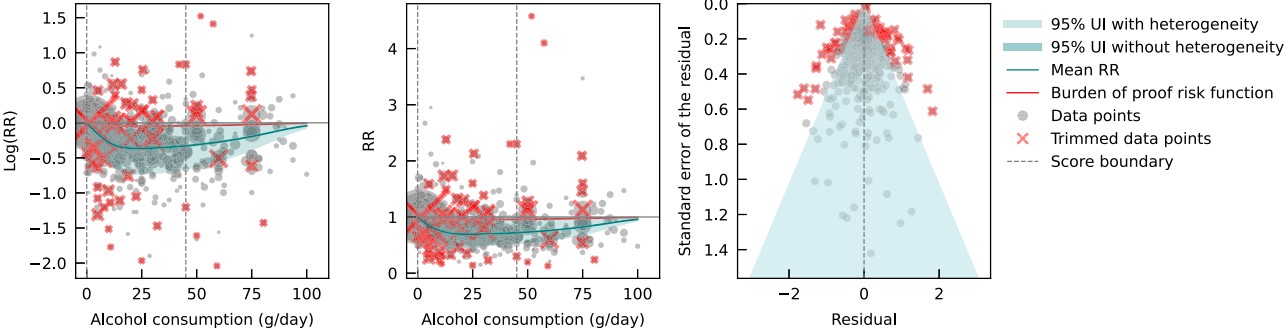

**Fig. 1 | Relative risk of alcohol consumption on ischemic heart disease, based on data from all conventional observational (cohort and case-control) studies.** The panels show the log(relative risk) function, the relative risk function, and a modified funnel plot showing the residuals (relative to 0) on the x-axis and the estimated standard error that includes the reported standard error and between-study heterogeneity on the y-axis. RR relative risk, UI uncertainty interval. Source data are provided as a Source Data file.

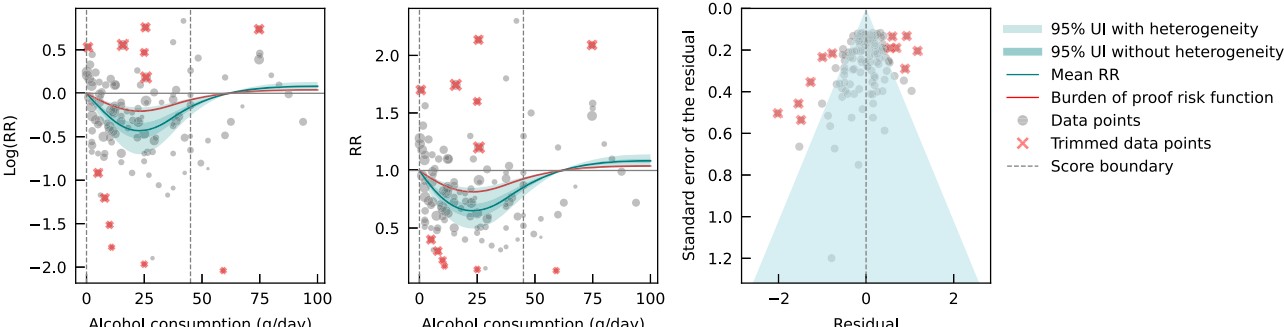

**Fig. 2 | Relative risk of alcohol consumption on ischemic heart disease, based on data from case-control studies.** The panels show the log(relative risk) function, the relative risk function, and a modified funnel plot showing the residuals (relative to 0) on the x-axis and the estimated standard deviation that includes the reported standard deviation and between-study heterogeneity on the y-axis. RR relative risk, UI uncertainty interval. Source data are provided as a Source Data file.

self-report data and those of other previous meta-analyses[4,35,158] that pooled conventional observational studies. Based on the conservative burden of proof interpretation of the data, our results suggested an inverse association between alcohol and IHD when all conventional observational studies were pooled (alcohol intake was associated with a reduction in IHD risk by an average of at least 4% across average consumption levels; two-star rating). In evaluating only cohort studies, we again found an inverse association between alcohol consumption and IHD (alcohol intake was associated with a reduction in IHD risk by an average of at least 5% at average consumption levels; two-star rating). In contrast, when we pooled only case-control studies, we estimated that average levels of alcohol consumption were associated with at least a 13% average decrease in IHD risk (three-star rating), but the inverse association reversed when consumption exceeded 60 g/day, suggesting that alcohol above this level is associated with a slight increase in IHD risk. Our analysis of the available evidence from MR studies showed no association between genetically predicted alcohol consumption and IHD.

Various potential biases and differences in study designs may have contributed to the conflicting findings. In our introduction, we summarized important sources of bias in conventional observational studies of the association between alcohol consumption and IHD. Of greatest concern are residual and unmeasured confounding and reverse causation, the effects of which are difficult to eliminate in conventional observational studies. By using SNPs within an IV approach to predict exposure, MR – in theory – eliminates these sources of bias and allows for more robust estimates of causal effects. Bias may still occur, however, when using MR to estimate the association between alcohol and IHD[159,160]. There is always the risk of horizontal pleiotropy in MR – that is, the genetic variant may affect the outcome via pathways other than exposure[161]. The IV assumption of exclusion restriction is, for example, violated if only a single measurement of alcohol consumption is used in MR[162]; because alcohol consumption varies over the life course, the gene directly impacts IHD through intake at time points other than that used in the MR analysis. To date, MR studies have not succeeded in separately capturing the multidimensional effects of alcohol intake on IHD risk (i.e., effects of average alcohol consumption measured through frequency-quantity, in addition to the effects of HED)[159] because the genes used to date only target average alcohol consumption that encompasses intake both at average consumption levels and HED. In other words, the instruments used are not able to separate out the individual effects of these two different dimensions of alcohol consumption on IHD risk using MR. Moreover, reverse causation may occur through cross-generational effects[160,163], as the same genetic variants predispose both the individual and at least one of his or her parents to (increased) alcohol consumption. In this situation, IHD risk could be associated with the parents' genetically predicted alcohol consumption and not with the individual's own consumption. None of the MR studies included accounted for cross-generational effects, which possibly introduced bias in the effect estimates. It is important to note that bias by ancestry might also occur in conventional observational studies[164]. In summary, estimates of the alcohol-IHD association are prone to bias in all three study designs, limiting inferences of causation.

The large difference in the number of available MR versus conventional observational studies, the substantially divergent results

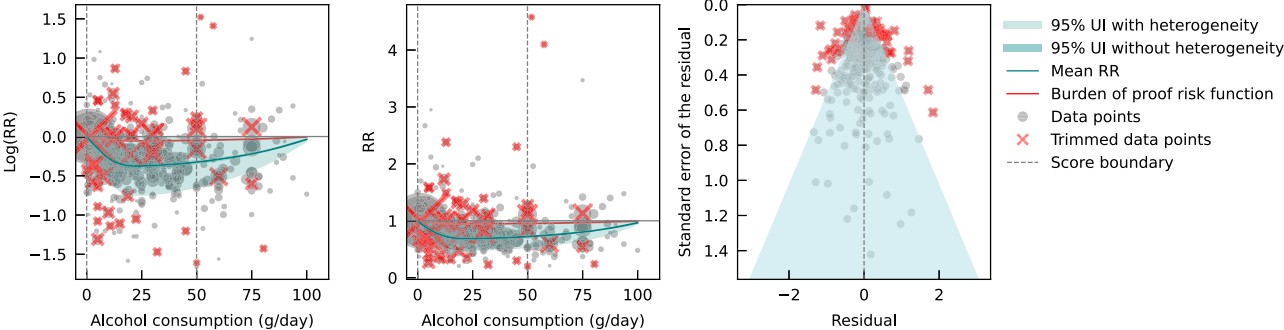

**Fig. 3 | Relative risk of alcohol consumption on ischemic heart disease, based on data from cohort studies.** The panels show the log(relative risk) function, the relative risk function, and a modified funnel plot showing the residuals (relative to 0) on the x-axis and the estimated standard deviation that includes the reported standard deviation and between-study heterogeneity on the y-axis. RR relative risk, UI uncertainty interval. Source data are provided as a Source Data file.

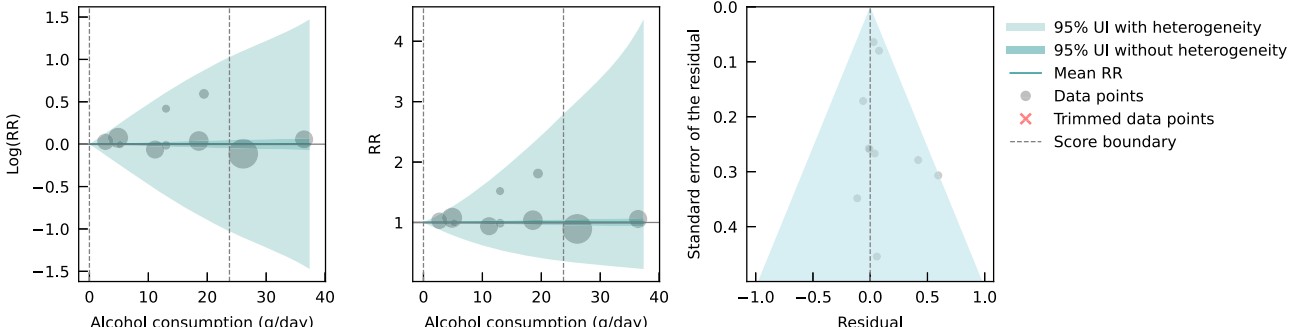

**Fig. 4 | Relative risk of alcohol consumption on ischemic heart disease, based on data from Mendelian randomization studies.** The panels show the log(relative risk) function, the relative risk function, and a modified funnel plot showing the residuals (relative to 0) on the x-axis and the estimated standard deviation that includes the reported standard deviation and between-study heterogeneity on the y-axis. RR relative risk, UI uncertainty interval. Source data are provided as a Source Data file.

derived from the different study types, and the rapidly developing field of MR clearly argue for further investigation of MR as a means to quantify the association between alcohol consumption and IHD risk. Future studies should investigate non-linearity in the relationship using non-linear MR methods. The residual method, commonly applied in non-linear MR studies such as Biddinger et al.[31], assumes a constant, linear relationship between the genetic IV and the exposure in the study population; a strong assumption that may result in biased estimates and inflated type I error rates if the relationship varies by population strata[165]. However, by log-transforming the exposure, the relationships between the genetic IV and the exposure as expressed on a logarithmic scale may be more homogeneous across strata, possibly reducing the bias effect of violating the assumption of a constant, linear relationship. Alternatively, or in conjunction, the recently developed doubly ranked method, which obviates the need for this assumption, could be used[166]. Since methodology for non-linear MR is an active field of study[167], potential limitations of currently available methods should be acknowledged and latest guidelines be followed[168]. Future MR studies should further (i) employ sensitivity analyses such as the MR weighted median method[169] to relax the exclusion restriction assumption that may be violated, as well as applying other methods such as the MR-Egger intercept test; (ii) use methods such as g-estimation of structural mean models[162] to adequately account for temporal variation in alcohol consumption in MR, and (iii) attempt to disaggregate the effects of alcohol on IHD by dimension in MR, potentially through the use of MVMR[164]. General recommendations to overcome common MR limitations are described in greater detail elsewhere[159,163,170,171] and should be carefully considered. With respect to prospective cohort studies used to assess the alcohol-IHD

relationship, they should, at a minimum: (i) adjust the association between alcohol consumption and IHD for all potential confounders identified, for example, using a causal directed acyclic graph, and (ii) account for reverse causation introduced by sick quitters and by drinkers who changed their consumption. If possible, they should also (iii) use alcohol biomarkers as objective measures of alcohol consumption instead of or in addition to self-reported consumption to reduce bias through measurement error, (iv) investigate the association between IHD and HED, in addition to average alcohol consumption, and (v) when multiple measures of alcohol consumption and potential confounders are available over time, use g-methods to reduce bias through confounding as fully as possible within the limitations of the study design. However, some bias – due, for instance, to unmeasured confounding in conventional observational and to horizontal pleiotropy in MR studies – is likely inevitable, and the interpretation of estimates should be appropriately cautious, in accordance with the methods used in the study.

With the introduction of the Moderate Alcohol and Cardiovascular Health Trial (MACH15)[172], randomized controlled trials (RCTs) have been revisited as a way to study the long-term effects of low to moderate alcohol consumption on cardiovascular disease, including IHD. In 2018, soon after the initiation of MACH15, the National Institutes of Health terminated funding[173], reportedly due to concerns about study design and irregularities in the development of funding opportunities[174]. Although MACH15 was terminated, its initiation represented a previously rarely considered step toward investigating the alcohol-IHD relationship using an RCT[175]. However, while the insights from an RCT are likely to be invaluable, the implementation is fraught with potential issues. Due to the growing

number of studies suggesting increased disease risk, including cancer[3,4], associated with alcohol use even at very low levels[176], the use of RCTs to study alcohol consumption is ethically questionable[177]. A less charged approach could include the emulation of target trials[178] using existing observational data (e.g., from large-scale prospective cohort studies such as the UK Biobank[179], Atherosclerosis Risk in Communities Study[180], or the Framingham Heart Study[181]) in lieu of real trials to gather evidence on the potential cardiovascular effects of alcohol. Trials like MACH15 can be emulated, following the proposed trial protocols as closely as the observational dataset used for the analysis allows. Safety and ethical concerns, such as those related to eligibility criteria, initiation/increase in consumption, and limited follow-up duration, will be eliminated because the data will have already been collected. This framework allows for hypothetical trials investigating ethically challenging or even untenable questions, such as the long-term effects of heavy (episodic) drinking on IHD risk, to be emulated and inferences to broader populations drawn.

There are several limitations that must be considered when interpreting our findings. First, record screening for our systematic review was not conducted in a double-blinded fashion. Second, we did not have sufficient evidence to estimate and examine potential differential associations of alcohol consumption with IHD risk by beverage type or with MI endpoints by sex. Third, despite using a flexible meta-regression tool that overcame several limitations common to meta-analyses, the results of our meta-analysis were only as good as the quality of the studies included. We were able, however, to address the issue of varying quality of input data by adjusting for bias covariates that corresponded to core study characteristics in our analyses. Fourth, because we were only able to include one-sample MR studies that captured genetically predicted alcohol consumption, statistical power may be lower than would have been possible with the inclusion of two-sample MR studies, and studies that directly estimated gene-IHD associations were not considered[23]. Finally, we were not able to account for participants' HED status when pooling effect size estimates from conventional observational studies. Given established differences in IHD risk for drinkers with and without HED[35] and the fact that more than one in three drinkers reports HED[6], we would expect that the decreased average risk we found at moderate levels of alcohol consumption would be attenuated (i.e., approach the IHD risk of non-drinkers) if the presence of HED was taken into account.

Using the burden of proof approach[32], we conservatively re-evaluated the dose-response relationship between alcohol consumption and IHD risk based on existing cohort, case-control, and MR data. Consistent with previous meta-analyses, we found that alcohol at average consumption levels was inversely associated with IHD when we pooled conventional observational studies. This finding was supported when aggregating: (i) all studies, (ii) only cohort studies, (iii) only case-control studies, (iv) studies examining IHD morbidity in females and males, (v) studies examining IHD mortality in males, and (vi) studies examining MI morbidity. In contrast, we found no association between genetically predicted alcohol consumption and IHD risk based on data from MR studies. Our confirmation of the conflicting results derived from self-reported versus genetically predicted alcohol use data highlights the need to advance methodologies that will provide more definitive answers to this critical public health question. Given the limitations of randomized trials, we advocate using advanced MR techniques and emulating target trials using observational data to generate more conclusive evidence on the long-term effects of alcohol consumption on IHD risk.

## Methods

This study was approved by the University of Washington IRB Committee (study #9060).

## Overview

The burden of proof approach is a six-step framework for conducting meta-analysis[32]: (1) data from published studies that quantified the dose-response relationship between alcohol consumption and ischemic heart disease (IHD) risk were systematically identified and obtained; (2) the shape of the mean relative risk (RR) curve (henceforth 'risk curve') and associated uncertainty was estimated using a quadratic spline and algorithmic trimming of outliers; (3) the risk curve was tested and adjusted for biases due to study attributes; (4) unexplained between-study heterogeneity was quantified, adjusting for within-study correlation and number of studies included; (5) the evidence for small-study effects was evaluated to identify potential risks of publication or reporting bias; and (6) the burden of proof risk function (BPRF) – a conservative interpretation of the average risk across the exposure range found in the data – was estimated relative to IHD risk at zero alcohol intake. The BPRF was converted to a risk-outcome score (ROS) that was mapped to a star rating from one to five to provide an intuitive interpretation of the magnitude and direction of the dose-response relationship between alcohol consumption and IHD risk.

We calculated the mean RR and 95% uncertainty intervals (UIs) for IHD associated with levels of alcohol consumption separately with all evidence available from conventional observational studies and from Mendelian randomization (MR) studies. For the risk curves that met the condition of statistical significance when the conventional 95% UI that does not include unexplained between-study heterogeneity was evaluated, we calculated the BPRF, ROS, and star rating. Based on input data from conventional observational studies, we also estimated these metrics by study design (cohort studies, case-control studies), and by IHD endpoint (morbidity, mortality) for both sexes (females, males) and sex-specific. For sex-stratified analyses, we only considered studies that reported effect sizes for both females and males to allow direct comparison of IHD risk across different exposure levels; however, we did not collect information about the method each study used to determine sex. We also estimated risk curves for myocardial infarction (MI), overall and by endpoint, using data from conventional observational studies. As a comparison, we also estimated each risk curve without trimming 10% of the input data. We did not consider MI as an outcome or disaggregate findings by sex or endpoint for MR studies due to insufficient data.

With respect to MR studies, several statistical methods are typically used to estimate the associations between genetically predicted exposure and health outcomes (e.g., two-stage least squares [2SLS], inverse-variance-weighted [IVW], multivariable Mendelian randomization [MVMR]). For our main analysis synthesizing evidence from MR studies, we included the reported effect sizes estimated using 2SLS if a study applied multiple methods because this method was common to all included studies. In sensitivity analyses, we used the effect sizes obtained by other MR methods (i.e., IVW, MVMR, and non-linear MR) and estimated the mean risk curve and uncertainty. We also pooled conventionally estimated effect sizes from MR studies to allow comparison with the risk curve estimated with cohort studies. Due to limited input data from MR studies, we elected not to trim 10% of the observations. Furthermore, we estimated the risk curve from cohort studies with data from countries that corresponded to those included in MR studies (China, the Republic of Korea, and the United Kingdom). Due to a lack of data, we were unable to estimate a risk curve from case-control studies in these geographic regions.

## Conducting the systematic review

In step one of the burden of proof approach, data for the dose-response relationship between alcohol consumption and IHD risk were systematically identified, reviewed, and extracted. We updated a previously published systematic review[1] in PubMed that identified all studies evaluating the dose-response relationship between alcohol consumption and risk of IHD morbidity or mortality from January 1,

1970, to December 31, 2019. In our update, we additionally considered all studies up to and including December 31, 2021, for eligibility. We searched articles in PubMed on March 21, 2022, with the following search string: (alcoholic beverage[MeSH Terms] OR drinking behavior[MeSH Terms] OR "alcohol"[Title/Abstract]) AND (Coronary Artery Disease[Mesh] OR Myocardial Ischemia[Mesh] OR atherosclerosis[Mesh] OR Coronary Artery Disease[TiAb] OR Myocardial Ischemia[TiAb] OR cardiac ischemia[TiAb] OR silent ischemia[TiAb] OR atherosclerosis Outdent [TiAb] OR Ischemic heart disease[TiAb] OR Ischemic heart disease[TiAb] OR coronary heart disease[TiAb] OR myocardial infarction[TiAb] OR heart attack[TiAb] OR heart infarction[TiAb]) AND (Risk[MeSH Terms] OR Odds Ratio[MeSH Terms] OR "risk"[Title/Abstract] OR "odds ratio"[Title/Abstract] OR "cross-product ratio"[Title/Abstract] OR "hazards ratio"[Title/Abstract] OR "hazard ratio"[Title/Abstract]) AND ("1970/01/01"[PDat]: "2021/12/31"[PDat]) AND (English[LA]) NOT (animals[MeSH Terms] NOT Humans[MeSH Terms]). Studies were eligible for inclusion if they met all of the following criteria: were published between January 1, 1970, and December 31, 2021; were a cohort study, case-control study, or MR study; described an association between alcohol consumption and IHD and reported an effect size estimate (relative risk, hazard ratio, odds ratio); and used a continuous dose as exposure of alcohol consumption. Studies were excluded if they met any of the following criteria: were an aggregate study (meta-analysis or pooled cohort); utilized a study design not designated for inclusion in this analysis: not a cohort study, case-control study, or MR study; were a duplicate study: the underlying sample of the study had also been analyzed elsewhere (we always considered the analysis with the longest follow-up for cohort studies or the most recently published analysis for MR studies); did not report on the exposure of interest: reported on combined exposure of alcohol and drug use or reported alcohol consumption in a non-continuous way; reported an outcome that was not IHD or a composite outcome that included but was not limited to IHD, or outcomes lacked specificity, such as cardiovascular disease or all-cause mortality; were not in English; and were animal studies. All screenings of titles and abstracts of identified records, as well as full texts of potentially eligible studies, and extraction of included studies, were done by a single reviewer (SC or HL) independently. If eligible, studies were extracted for study characteristics, exposure, outcome, adjusted confounders, and effect sizes and their uncertainty. While the previous systematic review only considered cohort and case-control studies, our update also included MR studies. We chose to consider only 'one-sample' MR studies, i.e., those in which genes, risk factors, and outcomes were measured in the same participants, and not 'two-sample' MR studies in which two different samples were used for the MR analysis so that we could fully capture study-specific information. We re-screened previously identified records for MR studies to consider all published MR studies in the defined time period. We also identified and included in our sensitivity analysis an MR study published in 2022[31] which used a non-linear MR method to estimate the association between genetically predicted alcohol consumption and IHD. When eligible studies reported both MR and conventionally estimated effect sizes (i.e., for the association between self-reported alcohol consumption and IHD risk), we extracted both. If studies used the same underlying sample and investigated the same outcome in the same strata, we included the study that had the longest follow-up. This did not apply when the same samples were used in conventional observational and MR studies, because they were treated separately when estimating the risk curve of alcohol consumption and IHD. Continuous exposure of alcohol consumption was defined as a frequency-quantity measure[182] and converted to g/day. IHD was defined according to the International Classification of Diseases (ICD) −9, 410-414, and ICD-10, I20-I25.

The raw data were extracted with a standardized extraction sheet (see Supplementary Information Section 3, Table S4). For conventional observational studies, when multiple effect sizes were estimated from differently adjusted regression models, we used those estimated with the model reported to be fully adjusted or the one with the most covariates. In the majority of studies, alcohol consumption was categorized based on the exposure range available in the data. If the lower end of a categorical exposure range (e.g., <10 g/day) of an effect size was not specified in the input data, we assumed that this was 0 g/day. If the upper end was not specified (e.g., >20 g/day), it was calculated by multiplying the lower end of the categorical exposure range by 1.5. When the association between alcohol and IHD risk was reported as a linear slope, the average consumption level in the sample was multiplied by the logarithm of the effect size to effectively render it categorical. From the MR study which employed non-linear MR[31], five effect sizes and their uncertainty were extracted at equal intervals across the reported range of alcohol exposure using WebPlotDigitizer. To account for the fact that these effect sizes were derived from the same non-linear risk curve, we adjusted the extracted standard errors by multiplying them by the square root of five (i.e., the number of extracted effect sizes). Details on data sources are provided in Supplementary Information Section 4.

## Estimating the shape of the risk-outcome relationship

In step two, the shape of the dose-response relationship (i.e., 'signal') between alcohol consumption and IHD risk was estimated relative to risk at zero alcohol intake. The meta-regression tool MR-BRT (meta-regression−Bayesian, regularized, trimmed), developed by Zheng et al.[33], was used for modeling. To allow for non-linearity, thus relaxing the common assumption of a log-linear relationship, a quadratic spline with two interior knots was used for estimating the risk curve[33]. We used the following three risk measures from included studies: RRs, odds ratios (ORs), and hazard ratios (HRs). ORs were treated as equivalent to RRs and HRs based on the rare outcome assumption. To counteract the potential influence of knot placement on the shape of the risk curve when using splines, an ensemble model approach was applied. Fifty component models with random knot placements across the exposure domain were computed. These were combined into an ensemble by weighting each model based on model fit and variation (i.e., smoothness of fit to the data). To prevent bias from outliers, a robust likelihood-based approach was applied to trim 10% of the observations. Technical details on estimating the risk curve, use of splines, the trimming procedure, the ensemble model approach, and uncertainty estimation are described elsewhere[32,33]. Details on the model specifications for each risk curve are provided in Supplementary Information section 8. We first estimated each risk curve without trimming input data to visualize the shape of the curve, which informed knot placement and whether to set a left and/or right linear tail when data were sparse at low or high exposure levels (see Supplementary Information Section 10, Fig. S5a–l).

## Testing and adjusting for biases across study designs and characteristics

In step three, the risk curve was tested and adjusted for systematic biases due to study attributes. According to the Grading of Recommendations Assessment, Development, and Evaluation (GRADE) criteria[183], the following six bias sources were quantified: representativeness of the study population, exposure assessment, outcome ascertainment, reverse causation, control for confounding, and selection bias. Representativeness was quantified by whether the study sample came from a location that was representative of the underlying geography. Exposure assessment was quantified by whether alcohol consumption was recorded once or more than once in conventional observational studies, or with only one or multiple SNPs in MR studies. Outcome ascertainment was quantified by whether IHD was ascertained by self-report only or by at least one other measurement method. Reverse causation was quantified by whether increased IHD

risk among participants who reduced or stopped drinking was accounted for (e.g., by separating former drinkers from lifetime abstainers). Control for confounding factors was quantified by which and how many covariates the effect sizes were adjusted for (i.e., through stratification, matching, weighting, or standardization). Because the most adjusted effect sizes in each study were extracted in the systematic review process and thus may have been adjusted for mediators, we additionally quantified a bias covariate for each of the following potential mediators of the alcohol-IHD relationship: body mass index, blood pressure, cholesterol (excluding high-density lipo-protein cholesterol), fibrinogen, apolipoprotein A1, and adiponectin. Selection bias was quantified by whether study participants were selected and included based on pre-existing disease states. We also quantified and considered as possible bias covariates whether the reference group was non-drinkers, including lifetime abstainers and former drinkers; whether the sample was under or over 50 years of age; whether IHD morbidity, mortality, or both endpoints were used; whether the outcome mapped to IHD or referred only to subtypes of IHD; whether the outcome mapped to MI; and what study design (cohort or case-control) was used when conventional observational studies were pooled. Details on quantified bias covariates for all included studies are provided in Supplementary Information section 5 (Tables S7 and S8). Using a Lasso approach[184], the bias covariates were first ranked. They were then included sequentially, based on their ranking, as effect modifiers of the 'signal' obtained in step two in a linear meta-regression. Significant bias covariates were included in modeling the final risk curve. Technical details of the Lasso procedure are described elsewhere[32].

### Quantifying between-study heterogeneity, accounting for heterogeneity, uncertainty, and small number of studies

In step four, the between-study heterogeneity was quantified, accounting for heterogeneity, uncertainty, and small number of studies. In a final linear mixed-effects model, the log RRs were regressed against the 'signal' and selected bias covariates, with a random intercept to account for within-study correlation and a study-specific random slope with respect to the 'signal' to account for between-study heterogeneity. A Fisher information matrix was used to estimate the uncertainty associated with between-study heterogeneity[185] because heterogeneity is easily underestimated or may be zero when only a small number of studies are available. We estimated the mean risk curve with a 95% UI that incorporated between-study heterogeneity, and we additionally estimated a 95% UI without between-study heterogeneity as done in conventional meta-regressions (see Supplementary Information Section 7, Table S10). The 95% UI incorporating between-study heterogeneity was calculated from the posterior uncertainty of the fixed effects (i.e., the 'signal' and selected bias covariates) and the 95% quantile of the between-study heterogeneity. The estimate of between-study heterogeneity and the estimate of the uncertainty of the between-study heterogeneity were used to determine the 95% quantile of the between-study heterogeneity. Technical details of quantifying uncertainty of between-study heterogeneity are described elsewhere[32].

### Evaluating potential for publication or reporting bias

In step five, the potential for publication or reporting bias was evaluated. The trimming algorithm used in step two helps protect against these biases, so risk curves found to have publication or reporting bias using the following methods were derived from data that still had bias even after trimming. Publication or reporting bias was evaluated using Egger's regression[34] and visual inspection using funnel plots. Egger's regression tested for a significant correlation between residuals of the RR estimates and their standard errors. Funnel plots showed the residuals of the risk curve against their standard errors. We reported publication or reporting bias when identified.

### Estimating the burden of proof risk function

In step six, the BPRF was calculated for risk-outcome relationships that were statistically significant when evaluating the conventional 95% UI without between-study heterogeneity. The BPRF is either the 5th (if harmful) or the 95th (if protective) quantile curve inclusive of between-study heterogeneity that is closest to the RR line at 1 (i.e., null); it indicates a conservative estimate of a harmful or protective association at each exposure level, based on the available evidence. The mean risk curve, 95% UIs (with and without between-study heterogeneity), and BPRF (where applicable) are visualized along with included effect sizes using the midpoint of each alternative exposure range (trimmed data points are marked with a red x), with alcohol consumption in g/day on the x-axis and (log)RR on the y-axis.

We calculated the ROS as the average log RR of the BPRF between the 15th and 85th percentiles of alcohol exposure observed in the study data. The ROS summarizes the association of the exposure with the health outcome in a single measure. A higher, positive ROS indicates a larger association, while a negative ROS indicates a weak association. The ROS is identical for protective and harmful risks since it is based on the magnitude of the log RR. For example, a mean log BPRF between the 15th and 85th percentiles of exposure of −0.6 (protective association) and a mean log BPRF of 0.6 (harmful association) would both correspond to a ROS of 0.6. The ROS was then translated into a star rating, representing a conservative interpretation of all available evidence. A star rating of 1 (ROS: <0) indicates weak evidence of an association, a star rating of 2 (ROS: 0–0.14) indicates a >0–15% increased or >0–13% decreased risk, a star rating of 3 (ROS: >0.14–0.41) indicates a >15–50% increased or >13–34% decreased risk, a star rating of 4 (ROS: >0.41–0.62) indicates a >50–85% increased or >34–46% decreased risk, and a star rating of 5 (ROS: >0.62) indicates a >85% increased or >46% decreased risk.

### Statistics & reproducibility

The statistical analyses conducted in this study are described above in detail. No statistical method was used to predetermine the sample size. When analyzing data from cohort and case-control studies, we excluded 10% of observations using a trimming algorithm; when analyzing data from MR studies, we did not exclude any observations. As all data used in this meta-analysis were from observational studies, no experiments were conducted, and no randomization or blinding took place.

### Reporting summary

Further information on research design is available in the Nature Portfolio Reporting Summary linked to this article.

## Data availability

The findings from this study were produced using data extracted from published literature. The relevant studies were identified through a systematic literature review and can all be accessed online as referenced in the current paper[26–29,31,39–157]. Further details on the relevant studies can be found on the GHDx website (https://ghdx.healthdata.org/record/ihme-data/gbd-alcohol-ihd-bop-risk-outcome-scores). Study characteristics of all relevant studies included in the analyses are also provided in Supplementary Information Section 4 (Tables S5 and S6). The template of the data collection form is provided in Supplementary Information section 3 (Table S4). The source data includes processed data from these studies that underlie our estimates. Source data are provided with this paper.

## Code availability

Analyses were carried out using R version 4.0.5 and Python version 3.10.9. All code used for these analyses is publicly available online (https://github.com/ihmeuw-msca/burden-of-proof).

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

## Acknowledgements

Research reported in this publication was supported by the Bill & Melinda Gates Foundation [OPP1152504]. S.L. has received grants or contracts from the UK Medical Research Council [MR/T017708/1], CDC Foundation [project number 996], World Health Organization [APW No 2021/1194512], and is affiliated with the NIHR Oxford Biomedical Research Centre. The University of Oxford's Clinical Trial Service Unit and Epidemiological Studies Unit (CTSU) is supported by core grants from the Medical Research Council [Clinical Trial Service Unit A310] and the British Heart Foundation [CH/1996001/9454]. The CTSU receives research grants from industry that are governed by University of Oxford contracts that protect its independence and has a staff policy of not taking personal payments from industry. The content is solely the responsibility of the authors and does not necessarily represent the official views of the funders. The funders of the study had no role in study design, data collection, data analysis, data interpretation, writing of the final report, or the decision to publish.

## Author contributions

S.C., S.A.M., S.I.H., and E.C.M. managed the estimation or publications process. S.C. wrote the first draft of the manuscript. S.C. had primary responsibility for applying analytical methods to produce estimates. S.C. and H.R.L. had primary responsibility for seeking, cataloging, extracting, or cleaning data; designing or coding figures and tables. S.C., D.B., S.B., E.C.M., S.I.N., J.R., and R.J.D.S. provided data or critical feedback on data sources. S.C., D.B., P.Z., A.Y.A., S.I.N., and R.J.D.S. developed methods or computational machinery. S.C., D.B., P.Z., S.B., S.I.H., E.C.M., C.J.L.M., S.I.N., J.R., R.J.D.S., S.L., and E.G. provided critical feedback on methods or results. S.C., D.B., S.A.M., S.B., S.I.H., C.J.L.M., J.R., G.A.R., S.L., and E.G. drafted the work or revised it critically for important intellectual content. S.C., S.I.H., E.C.M., and E.G. managed the overall research enterprise.

## Competing interests

G.A.R. has received support for this manuscript from the Bill and Melinda Gates Foundation [OPP1152504]. S.L. has received grants or contracts from the UK Medical Research Council [MR/T017708/1], CDC Foundation [project number 996], World Health Organization [APW No 2021/1194512], and is affiliated with the NIHR Oxford Biomedical Research Centre. The University of Oxford's Clinical Trial Service Unit and Epidemiological Studies Unit (CTSU) is supported by core grants from the Medical Research Council [Clinical Trial Service Unit A310] and the British Heart Foundation [CH/1996001/9454]. The CTSU receives research grants from industry that are governed by University of Oxford contracts that protect its independence and has a staff policy of not taking personal payments from industry. All other authors declare no competing interests.
