## [Peer Review File · Nature Communications]

REVIEWERS' COMMENTS

Reviewer #1 (Remarks to the Author):

The question of whether alcohol consumption has a beneficial effect on cardiovascular diseases has been a topic of ongoing debate within the scientific community and many Mendelian randomisation (MR) studies have tried to address it. Recently, a review article collected 25 studies to rule out any positive influence of alcohol consumption on cardiovascular health (<https://link.springer.com/article/10.1007/s10654-021-00799-5>). I do not agree with the authors who stated that MR studies are non-conclusive due to biases such as pleiotropy. Negative MR results are more conclusive than positive results and pleiotropy is unlikely to produce spurious negative results. As an example, C-reactive protein was debated as a potential causal factor for cardiovascular diseases in the late 1990s and early 2000s, however, their potential causal role was ruled out when it was shown that genetically elevated CRP is not associated with the risk of cardiovascular diseases.

However, it is essential to acknowledge that current MR methods may not fully address the non-linear relationship between alcohol consumption and cardiovascular outcomes which has been reported by observational studies. The evolving field of MR offers hope for more advanced methods to address this limitation and I hope that we can soon reach a conclusion using the MR framework in an observational setting. In light of this possibility, I agree with the reviewer that running clinical trials is not ethically possible and not highly necessary.

Reviewer #2 (Remarks to the Author):

Thank you for the opportunity to review a well-written manuscript summarising the observational and genetic evidence on the association between alcohol consumption and ischaemic heart disease. I agree with the authors' response to the first and fourth reviewer that publishing their article on Nature Communications would create momentum around the relationship of alcohol and ischemic heart disease.

One concern I have is about the pooling of estimates from MR studies - I notice that two of them (Biddinger et al. and Lankester et al.) are carried out in the same dataset. Did the authors account for the correlation between estimates from these studies?

Additionally, I have to say that I agree with reviewer comments to include only either logRR or RR results. I believe that the latter approach would enable greater accessibility to a larger audience.

I wonder also whether it would be possible to have the results for myocardial infarction stratified by sex in Table 2, consistently with those for ischemic heart disease.

Minor comments:

Lines 69-70: "With effects likely to vary materially by patterns of drinking, alcohol consumption must be considered a multidimensional risk factor for IHD." is contradicted by a previous sentence that summarises the protective effects of alcohol intake. I suggest rephrasing "risk factor" for consistency with this previous statement.

Line 158: The frequency of records in the manuscript (4,826) do not seem in agreement with Extended Data Figure 1 (PRISMA diagram). Also, the sum of included studies (97+26+4) is different from the one in the diagram (126).

Reviewer #3 (Remarks to the Author):

Thank you for addressing my comments., which are very thoughtful and comprehensive. A few comments remained which may improve the overall paper.

Minor comments

- Thank you for incorporating the findings from non-linear MR analyses. However, with recent development in the issues possibly biasing non-linear MR (which appears after my initial review), the authors could also mention the possible bias in these non-linear MR in the Discussion ([https://www.thelancet.com/journals/landia/article/PIIS2213-8587\(23\)00198-5/fulltext](https://www.thelancet.com/journals/landia/article/PIIS2213-8587(23)00198-5/fulltext)). I personally think that this does not automatically trash the analyses from the study but just need to be aware of possible biases to cover all grounds.

- I understand the rationale for using one-sample MR versus two-sample MR analyses. However, omitting the latter may risk only including studies with lower statistical power. I do not have strong preference on this but perhaps the impact should be mentioned in the Methods/limitations

Discretionary comment

- The use of RCTs to address alcohol and health has been proposed in the past (e.g. MACH) although the study was terminated due to possible conflict of interest. With increasing number of studies suggesting harms of alcohol even at very low levels, as commented by WHO ([https://www.thelancet.com/journals/lanpub/article/PIIS2468-2667\(22\)00317-6/fulltext](https://www.thelancet.com/journals/lanpub/article/PIIS2468-2667(22)00317-6/fulltext)) and hence advocating trials against this background may be problematic ([https://www.thelancet.com/journals/lancet/article/PIIS0140-6736\(18\)32214-1/fulltext](https://www.thelancet.com/journals/lancet/article/PIIS0140-6736(18)32214-1/fulltext))?

Reviewer #1

Remarks to the Author:

The question of whether alcohol consumption has a beneficial effect on cardiovascular diseases has been a topic of ongoing debate within the scientific community and many Mendelian randomisation (MR) studies have tried to address it. Recently, a review article collected 25 studies to rule out any positive influence of alcohol consumption on cardiovascular health (<https://link.springer.com/article/10.1007/s10654-021-00799-5>). I do not agree with the authors who stated that MR studies are non-conclusive due to biases such as pleiotropy. Negative MR results are more conclusive than positive results and pleiotropy is unlikely to produce spurious negative results. As an example, C-reactive protein was debated as a potential causal factor for cardiovascular diseases in the late 1990s and early 2000s, however, their potential causal role was ruled out when it was shown that genetically elevated CRP is not associated with the risk of cardiovascular diseases.

However, it is essential to acknowledge that current MR methods may not fully address the non-linear relationship between alcohol consumption and cardiovascular outcomes which has been reported by observational studies. The evolving field of MR offers hope for more advanced methods to address this limitation and I hope that we can soon reach a conclusion using the MR framework in an observational setting. In light of this possibility, I agree with the reviewer that running clinical trials is not ethically possible and not highly necessary.

Response 1:

We thank the reviewer for sharing their concern regarding our summary of the current evidence on the association between alcohol and IHD using MR, including the potential issues we highlighted. We agree that negative results observed in MR are more conclusive than positive results. However, we do not agree that pleiotropy is unlikely to produce spurious negative results observed in some MR studies if the true causal effect of alcohol on IHD is harmful or protective. If, for example, the genetic variant affects any (or several) of the confounders of the alcohol-IHD relationship which bias the association toward the null, then the MR estimate would also be biased toward the null. In addition to the ever-present risk of horizontal pleiotropy when using an instrumental variable and potential pleiotropy by affecting confounders, there are other reasons why estimates of the association between alcohol and IHD may be biased (towards the null) in MR. These include, for example, reverse causality due to cross-generational effects or violations of the exclusion restriction assumption when only a single measure of alcohol consumption is used and thus temporal variations in alcohol consumption are not taken into account, introducing horizontal pleiotropy (Shi et al., 2022, proposed methodological solution to this problem using g-estimation). We are not aware of any MR study of alcohol consumption and ischemic heart disease that addressed either of these potential biases and we look forward to seeing future research accounting for these sources of bias. In summary, several sources of bias could have contributed to the observed null finding. It is important to recognize these potential biases when using MR and, if possible, take them into account. We think that a comprehensive and balanced discussion of the potential limitations of MR in studying the alcohol-IHD relationship is important and – by providing select recommendations on how to overcome them – may help future research efforts to mitigate bias in their analyses. We firmly believe that the developing field of MR offers the opportunity to investigate the relationship between alcohol consumption and IHD, and other cardiovascular diseases, more thoroughly and reliably. The triangulation of

future MR studies that adequately take into account potential bias and rigorous observational analyses (e.g., emulations of hypothetical randomized trials) will be instrumental in reaching consensus and answering this critical public health question.

We have revisited our discussion on the application of randomized trials to studying alcohol consumption and removed any recommendation to conduct a trial from our manuscript. Given that the University of Navarra Alumni Trialists Initiative (UNATI) – a four-year non-inferiority randomized trial to study the impact of alcohol consumption on major disease and mortality – was approved earlier this year, we elected to include a brief description of this trial in our discussion section. While acknowledging that randomized trials for alcohol use are ethically questionable and difficult to implement, in light of the recent approval of UNATI, we also believe it is important to acknowledge that randomized trials have the potential to provide key insights into the alcohol-in-moderation debate and cannot be completely disregarded. We hope our discussion is balanced and clearly delineates the ethical problems inherent in randomized trials, while also acknowledging the reality that a randomized trial to study the health effects of alcohol consumption has recently been approved and appears to be going forward.

“With the introduction of the Moderate Alcohol and Cardiovascular Health Trial (MACH15)¹⁶⁹, randomized controlled trials (RCTs) have been revisited as a way to study the long-term effects of low to moderate alcohol consumption on cardiovascular disease, including IHD. In 2018, soon after the initiation of MACH15, the National Institutes of Health terminated funding¹⁷⁰, reportedly due to concerns about study design and irregularities in the development of funding opportunities¹⁷¹. Although MACH15 was terminated, its initiation represented a previously rarely considered step toward investigating the alcohol-IHD relationship using an RCT¹⁷². In early 2023, funding was approved for the University of Navarra Alumni Trialists Initiative (UNATI)¹⁷³, a four-year non-inferiority RCT to study the impact of alcohol consumption on major disease and mortality. Participants will be women aged 55-70 and men aged 50-70 who already consume 3 to <40 drinks/week. They will be randomized into two groups, either i) women consuming up to 7 drinks/week and men consuming up to 14 drinks/week, while avoiding binge drinking, or ii) abstaining from alcohol. Participants will not be asked to initiate consumption or to increase their current levels of consumption. While the insights from an RCT would be invaluable, the implementation is fraught with potential issues. Due to the growing number of studies suggesting increased disease risk, including cancer^{3,4}, associated with alcohol use even at very low levels¹⁷⁴, the use of RCTs to study alcohol consumption is ethically questionable. A potential solution, however, could be the emulation of target trials¹⁷⁵ using existing observational data (e.g., from large-scale prospective cohort studies such as the UK Biobank¹⁷⁶, Atherosclerosis Risk in Communities Study¹⁷⁷, or the Framingham Heart Study¹⁷⁸) in lieu of – or complementary to – real trials to gather evidence on the potential cardiovascular effects of alcohol. Trials like MACH15 and UNATI can be emulated, following the proposed trial protocols as closely as the observational dataset used for the analysis allows. Safety and ethical concerns, such as those related to eligibility criteria, initiation/increase in consumption, and limited follow-up duration, will be eliminated because the data will have already been collected. This framework allows for hypothetical trials investigating ethically challenging or even untenable questions, such as the long-term effects of heavy (episodic) drinking on IHD risk, to be emulated and inferences to broader populations drawn.” (lines 352-377)

Reviewer #2

Remarks to the Author:

Thank you for the opportunity to review a well-written manuscript summarising the observational and genetic evidence on the association between alcohol consumption and ischaemic heart disease. I agree with the authors' response to the first and fourth reviewer that publishing their article on Nature Communications would create momentum around the relationship of alcohol and ischemic heart disease.

Response 2:

We are grateful for the reviewers' feedback on our manuscript and are pleased to hear that they agree with our view of creating momentum around this important public health issue by publishing with *Nature Communications*.

Comment:

One concern I have is about the pooling of estimates from MR studies - I notice that two of them (Biddinger et al. and Lankester et al.) are carried out in the same dataset. Did the authors account for the correlation between estimates from these studies?

Response 3:

We would like to thank the reviewer for recognizing the potential problem of modeling correlated data. Across all analyses we conducted (i.e. with data from cohort, case-control, and MR studies, respectively), we considered each underlying study (e.g., UK Biobank) only once to avoid – as the reviewer noted – study samples being considered more than once, which would need to be accounted for by statistical adjustment. We described this in the Methods section as follows:

“[studies were excluded if they] were a duplicate study: the underlying sample of the study had also been analyzed elsewhere (we always considered the analysis with the longest follow-up for cohort studies or the most recently published analysis for Mendelian randomization studies)”
(lines 473-475)

“If studies used the same underlying sample and investigated the same outcome in the same strata, we included the study that had the longest follow-up.” (lines 492-494).

Since both MR studies reported data from the UK Biobank, we only used the results of Biddinger et al. in the sensitivity analysis and therefore not the results of Lankester et al. (which we included in the main analysis) to avoid such “double counting” of data from the same study. We detailed this accordingly in the Methods section:

“We also used effect sizes from Biddinger and colleagues³² obtained using non-linear MR instead of those from Lankester and colleagues²⁹ in our main model (both were estimated with UK biobank data) to estimate a risk curve.” (line 269-271)

We hope this addresses the reviewer's concerns.

Comment:

Additionally, I have to say that I agree with reviewer comments to include only either logRR or RR results. I believe that the latter approach would enable greater accessibility to a larger audience.

Response 4:

We thank the reviewer for their comment and the recommendation to present only RR results. As this study is part of the burden of proof studies published serially in *Nature* family journals, we are keen to present the results consistently, including the visualization of the results. If there is a strong editorial direction to accommodate this revision, we will of course happily change it.

Comment:

I wonder also whether it would be possible to have the results for myocardial infarction stratified by sex in Table 2, consistently with those for ischemic heart disease.

Response 5:

We fully agree with the reviewers' suggestions that the analyses for the endpoints of myocardial infarction (morbidity and mortality) should be further stratified by sex to investigate effect modification. To reliably estimate sex-specific associations between alcohol consumption and the risk of morbidity and mortality from IHD, we only considered studies that reported effect sizes for both females and males to allow direct comparison of IHD morbidity and mortality risk across different exposure levels. However, there were unfortunately too few studies that investigated the risk of morbidity and mortality of myocardial infarction in relation to alcohol consumption levels separately for females and males. We thus have added this as a limitation of our study:

“Second, we did not have sufficient evidence to estimate and examine potential differential associations of alcohol consumption with IHD risk by beverage type or with MI endpoints by sex.” (lines 379-381)

Minor comments:**Comment:**

Lines 69-70: "With effects likely to vary materially by patterns of drinking, alcohol consumption must be considered a multidimensional risk factor for IHD." is contradicted by a previous sentence that summarises the protective effects of alcohol intake. I suggest rephrasing "risk factor" for consistency with this previous statement.

Response 6:

We thank the reviewer for pointing out this semantic detail. We have replaced the term "risk factor" with "factor" and reworded the sentence slightly:

“With effects likely to vary materially by patterns of drinking, alcohol consumption must be considered a multidimensional factor impacting IHD outcomes.” (lines 64-65)

Comment:

Line 158: The frequency of records in the manuscript (4,826) do not seem in agreement with Extended Data Figure 1 (PRISMA diagram). Also, the sum of included studies (97+26+4) is different from the one in the diagram (126).

Response 7:

We would like to thank the reviewer for their careful assessment and attention to detail. The frequency of records (4,826) reported in the text is the sum of records identified from databases and registers (4,769; shown in the center “column” of the PRISMA diagram) and records identified by other methods (57; shown in the right “column”). We have added this to the Results as follows to avoid any confusion:

“Of 4,826 records identified in our updated systematic review (4,769 from databases/registers and 57 by citation search and known literature), 11 were eligible based on our inclusion criteria and were included.” (lines 154-156)

The sum of cohort, case-control, and Mendelian randomization studies is higher than the total number of included studies because the four MR studies included in our main analysis also assessed the alcohol-IHD association using conventional observational methods. These studies were considered as cohort studies because their exact underlying data were not used in the included cohort studies. We have added the following note to the PRISMA to clarify this:

“In total, 97 cohort studies, 26 case-control studies, and five Mendelian randomization (MR) studies were included. All MR studies reported effect size estimates obtained using conventional methods from cohort data. As these exact underlying data were not used in the included cohort studies, we also included the four MR studies from our main analyses as cohort studies.”

Lastly, we found a typo in the number of studies from the previous systematic review, which we have corrected in the PRISMA diagram. In total, we considered data from 97 cohort studies, including 4 that are also MR studies, 26 case-control studies, and one additional MR study, totaling 124 (instead of 126). We very much appreciate the reviewer for bringing our attention to this mistake.

Reviewer #3

Remarks to the Author:

Thank you for addressing my comments, which are very thoughtful and comprehensive. A few comments remained which may improve the overall paper.

Response 8:

We are grateful to hear that we were able to address the reviewer's comments.

Minor comments:

Comment:

- Thank you for incorporating the findings from non-linear MR analyses. However, with recent development in the issues possibly biasing non-linear MR (which appears after my initial review), the authors could also mention the possible bias in these non-linear MR in the Discussion ([https://www.thelancet.com/journals/landia/article/PIIS2213-8587\(23\)00198-5/fulltext](https://www.thelancet.com/journals/landia/article/PIIS2213-8587(23)00198-5/fulltext)). I personally think that this does not automatically trash the analyses from the study but just need to be aware of possible biases to cover all grounds.

Response 9:

We thank the reviewer very much for drawing our attention to possible biases in the use of non-linear MR. We decided to highlight this limitation in the Discussion section when making recommendations on how future MR studies could be conducted to investigate the alcohol-IHD relationship. Specifically, when applying non-linear MR, we recommend using the doubly ranked method (instead of the residual method) to relax the assumption of constant, linear effects in population strata as described by Burgess (2023):

“Future studies should i) investigate non-linearity in the relationship, using non-linear MR methods such as the doubly ranked method, which obviates the need to assume constant, linear effects in population strata¹⁶⁵” (lines 331-333)

- I understand the rationale for using one-sample MR versus two-sample MR analyses. However, omitting the latter may risk only including studies with lower statistical power. I do not have strong preference on this but perhaps the impact should be mentioned in the Methods/limitations

Response 10:

We thank the reviewer for the comment. We agree and have added this to our limitations:

“Fourth, because we were only able to include one-sample MR studies that captured genetically predicted alcohol consumption, statistical power may be lower than would have been possible with the inclusion of two-sample MR studies, and studies that directly estimated gene-IHD associations were not considered²⁴.” (lines 385-388)

Discretionary comment:

- The use of RCTs to address alcohol and health has been proposed in the past (e.g. MACH) although the study was terminated due to possible conflict of interest. With increasing number of studies suggesting harms of alcohol even at very low levels, as commented by WHO ([https://www.thelancet.com/journals/lanpub/article/PIIS2468-2667\(22\)00317-6/fulltext](https://www.thelancet.com/journals/lanpub/article/PIIS2468-2667(22)00317-6/fulltext)) and hence advocating trials against this background may be problematic ([https://www.thelancet.com/journals/lancet/article/PIIS0140-6736\(18\)32214-1/fulltext](https://www.thelancet.com/journals/lancet/article/PIIS0140-6736(18)32214-1/fulltext))?

Response 11:

We thank the reviewer for sharing their concerns about advocating for trials for alcohol use. We have revisited our discussion on the application of randomized trials to studying alcohol consumption and removed any recommendations to conduct a trial from our manuscript. We have referenced the WHO comment shared by the reviewer in discussing the potential harms of alcohol as a key ethical challenge of trials. Given that the University of Navarra Alumni Trialists Initiative (UNATI) – a four-year non-inferiority randomized trial to study the impact of alcohol consumption on major disease and mortality – was approved earlier this year, we elected to include a brief description of this trial in our discussion section (please see **Response 1** for the full paragraph). While acknowledging that randomized trials for alcohol use are ethically questionable and difficult to implement, in light of the recent approval of the UNATI trial, we also believe it is important to acknowledge that randomized trials have the potential to provide key insights into the alcohol-in-moderation debate and cannot be completely disregarded. We hope our discussion is balanced and clearly delineates the ethical problems inherent in randomized trials, while also acknowledging the reality that a randomized trial to study the health effects of alcohol consumption has recently been approved and appears to be going forward.